# Effects of Climate and Land Use on the Population Dynamics of the Bank Vole (*Clethrionomys glareolus*) in the Southernmost Part of Its Range

**DOI:** 10.3390/ani15060839

**Published:** 2025-03-14

**Authors:** Lucía de la Huerta-Schliemann, Marc Vilella, Lídia Freixas, Ignasi Torre

**Affiliations:** 1Chrono-Environnement-UMR 6249 CNRS, Université de Franche-Comté, 16 route de Gray, Cedex, 25030 Besançon, France; 2Small Mammal Research Area and BiBio Research Group, Natural Sciences Museum of Granollers, Av. Francesc Macià 51, 08402 Granollers, Spain; marc.vilella16@gmail.com (M.V.); lfreixas@mcng.cat (L.F.); itorre@mcng.cat (I.T.)

**Keywords:** bank vole, population dynamics, climate change, land-use change, occupancy

## Abstract

Understanding how climate and land-use changes affect wildlife is essential for preserving biodiversity, especially in regions located at the limits of species’ natural ranges. This study focused on the bank vole, a typical northern small mammal species adapted to cooler climates that can be found in the forests of the Pyrenean and Mediterranean regions of Catalonia, in northeastern Spain. We wanted to know how weather and land use variables have influenced the population and occupancy dynamics of this species in Catalonia in the last 16 years. Thanks to the data available from the SEMICE monitoring program (monitoring of common small mammals in Spain), we were able to observe how the increase in temperature and the unpredictability of precipitation reduced the number of bank voles, especially in areas dominated by coniferous forests. Deciduous forests with richer vegetation covers were more favorable habitats. These results demonstrate that the combination of global warming and changes in land use could threaten bank vole populations in regions bordering the species’ range, thus highlighting the importance of favoring diverse forest habitats to help wildlife adapt to a changing climate.

## 1. Introduction

Climate and land-use change are among the main threats to biodiversity [1]. While climate-predicted scenarios have been available for some time, land-use predictions are more recent and potentially more complex [2]. Land-use models at the European scale have projected a 20–40% increase in cropland abandonment [3], consequently leading to an increase in forest areas in the continent, as has been reported in regions of Mediterranean Europe [4]. While this afforestation process can be beneficial for forest species, it implies the loss of many other important ecosystems and their associated species (e.g., pastures, scrublands, etc.). Conversely, recent climate projections suggest an increase in air temperature between 2 and 6 °C (e.g., [5,6]). Such temperature increases may not only affect the extent and quality of forests [7], but may also increase the risk of wildfires in previously fire-free areas, such as the Pyrenees in southern Europe [8]. The combined effects of climatic change and land shifts are likely to intensify the negative impacts on forest ecoregions in the coming decades, further threatening their biodiversity [9].

Focusing on the Mediterranean basin of the Iberian Peninsula, this region is characterized by a unique combination of biodiversity and environmental gradients. Catalonia, located in the northeast, exemplifies this diversity, ranging from the Atlantic-influenced humid areas of the Aran Valley to the subarid continental Mediterranean climate of the Ebro River plain [10]. The Mediterranean region dominates the lowlands, while the Eurosiberian zone includes most of the Pyrenees, Pre-Pyrenees, and coastal mountains, and the boreal alpine region is restricted to the higher altitudes [10,11]. Being subalpine and alpine systems under Mediterranean influence, most of their species are glacial remnants, nowadays at the limit of their climatic tolerance [12]. These transitions, driven by a mixture of climatic, geological, and historical influences, highlight the strong contrasts within relatively small geographic areas, making the Catalan Mediterranean basin a critical area for understanding species responses to environmental change.

In the Catalan mountains, the effects of climate change are especially intense due to the previously mentioned marginal conditions [13]. Regarding climate change in the axial Pyrenees, the summer precipitation has shown a slight decreasing trend, but winter snow accumulation is not expected to increase as a potential source of water in summer [14]. Predictions for the Catalan region are not expected to be better, foreseeing an increase in warming accompanied by a decrease in precipitation [15]. Climate and landscape changes can produce a significant reduction in range extent and habitat suitability for the most drought-sensitive forest species [16]. The modified environmental conditions will affect reproductive and survival rates, which can lead to local extinction or colonization. In the end, such changes in the population will lead to changes in the occurrence and the abundance of species across the landscape [17]. Understanding how these dynamic processes are affected by habitat or climate changes can be important for the management of ecological systems [18].

The bank vole (*Clethrionomys glareolus*) is a common small mammal of the Cricetidae family widely distributed in Europe and parts of Asia [19]. It is an epigeal forest-dwelling species with mid-European requirements, and is able to exploit a wide variety of food resources [20]. Although it can be found in open habitats like scrublands [21], grasslands [22], and agricultural landscapes [23], bank voles are dominant among the forest small rodents in Central and Northern Europe [24,25]. As it is a northern species adapted to cold climates [26], the Iberian Peninsula is one of the southernmost areas where this species can be found [27]. From an ecological point of view, bank voles play several important roles in the ecosystems; they form the main component of the diet of some forest aerial predators such as the tawny owl (*Strix aluco*) in Central Europe [28]. However, its contribution to the diet of open-habitat raptors (e.g., barn owl (*Tyto alba*) declines, especially towards the southernmost areas of the distribution range (NE Spain, [29]). In the latter area, bank voles are linked with forested and rocky habitats, and show significant penetrations in humid Mediterranean forests [30].

Bank voles help in the recycling and redistribution of nutrients in the ecosystems in which they live through their feeding behavior. In the Iberian Peninsula they do not have a negative economic impact [26] but it should be noted that they are a vector species of certain diseases that can affect humans (i.e., Hantavirus, [31]). Similarly to several species with northern requirements, the bank vole is expected to experience important range contractions under the new climate change projections during this century [32]. However, the role of other relevant factors (e.g., land use) was not considered since these predictions were exclusively modeled on climate envelopes, based on the assumption that climate mostly determines the distribution of a species [33]. Nonetheless, due to its known forest preferences, the conservation of its populations could depend to a large extent on the natural maintenance of forests, but this may not be a major concern in the context of the documented afforestation process in the Mediterranean region [4]. Indeed, the process of land abandonment during the last century in Mediterranean areas (especially in the mountains) may have increased habitat suitability for forest-dwelling species due to habitat gains by natural afforestation [34].

On the other hand, global warming is expected to have negative effects on species living in colder regions and habitats, and positive effects on those living in drier regions and environments [35]. Although these results were found in studies focused on passerine birds, a similar effect can be expected in small mammals, since they are homeotherms of similar size. In mountain areas, northern species are expected to shift upslope to counteract the increase in temperatures [36]. However, it is also clear that climate change is not the primary driver of the restructuring of the small mammal fauna in mountain areas [17]. Indeed, the recovery of forest cover after land-use changes over the past decades will help mitigate the predicted biological effects of environmental climatic change on small mammals [17]. As a forest-dwelling species adapted to cool and humid climates, bank voles could be favored by the afforestation process in its southernmost range [37] as was already depicted in several woodland bird species of northern climates [35]. Indeed, mountain areas will likely remain a stronghold for most of the cold-dwelling species in Catalonia [38].

In this study, we analyzed how climate and habitat affected several population parameters such as the abundance, population structure (e.g., age classes), and occupancy (and derived parameters) of the bank vole in Catalonia (NE Spain) at the southern limit of distribution in western Europe. We hypothesized that, under the global warming scenario predicted in Europe (i.e., increased temperatures, reduced precipitation, and afforestation), climate change would drive the retraction of bank vole populations, while landscape change might simultaneously facilitate their expansion by increasing habitat availability. Moreover, in the context of meta-population dynamics, we anticipated that population parameters such as occupancy probability would be conditioned by microhabitat characteristics (e.g., understorey vegetation), while local colonization and extinction would be influenced by the surrounding landscape matrix [39].

## 2. Materials and Methods

### 2.1. Study Area and Sampling Design

The study was conducted in Catalonia (NE Spain), encompassing a range of elevations from low Mediterranean coastal areas covered by evergreen sclerophyllous forests to high altitude coniferous forests and scree areas in the axial Pyrenees. This represents a continuous area where the bank vole is known to occur (Figure 1) according to the Atlas and Red book of the mammals of Spain [40]. This resource comprises a 10 × 10 km UTM grid across the entire territory and contains the information of occurrence of the bank vole.

Data were collected between spring 2008 and autumn 2023 by the SEMICE small mammals monitoring scheme (www.semice.org), a volunteer-based monitoring program designed to ensure adequate detectability of common species by reducing sampling bias [41]. This program consists of two sampling sessions per year, in spring and early summer (April to June), and autumn and early winter (October to December) [29]. The sampling method includes 36 traps on grids (6 × 6 trapping scheme) spaced 15 m apart, alternating in position 18 Sherman traps (folding Sherman trap for small animals; 23 × 7.5 × 9 cm, single rectangular piece of aluminum; Sherman Co., Tallahassee, FL, USA) with 18 Longworth traps (two pieces of aluminum: tunnel and nest, Penlon Ltd., Oxford, UK) [42,43] per sampling station. The traps operated for three consecutive nights and were baited (apple and a mixture of tuna, flour and oil) and provided with hydrophobic cotton for bedding and insulation [44]. In the Mediterranean zone, traps were checked daily in the morning (three checks), and in the Pyrenean zone; checks also included the two intermediate nights (five checks) to reduce overnight mortality due to more extreme weather conditions [45,46]. These two regions were selected according to the strong differences regarding air temperature, rainfall, extreme climatic events and seasonality [47], but also regarding biotic features [48], which were expected to affect the breeding period and the population dynamics of bank voles [26,49]. In this regard, Catalonia showed a biotic boundary separating the Pyrenees from the remainder area [48].

The selection criteria consisted of those available SEMICE stations that recorded the presence of the species and that had been active for at least four sampling sessions. This resulted in a dataset comprising 29 stations and 32 trapping sessions over a 16-year period. The sampling plots that met these requirements were located along a strong elevation gradient (200–2066 m a.s.l.) and mainly within the Natura 2000 protected areas of Catalonia.

Captured bank voles were identified, weighed, sexed and ear-tagged (small-animal tag, National Band Co., Newport, KY, USA) and released at the point of capture [50]. Research with live animals followed ethical guidelines [51], and captures were performed under special permission from the government of Catalonia (Generalitat de Catalunya).

Climatic data were requested from the Catalan meteorological service [52] as they have recently developed monthly interpolated maps of temperatures (mean) and precipitation (accumulated) based on weather stations since January 2008. Once the maps were obtained, the values of these two variables for each sampling station for the whole time series were extracted using the Raster package [53]. Changes between periods were evaluated using linear regression models. We also calculated the water deficit (HD = precipitation-potential evapotranspiration, using the Thornwhite methodology [54,55] as a reference of drought conditions. The mean temperature, as well as the cumulative precipitation of the three, six, and twelve months prior to each sampling period were considered (see [56,57] for a similar approach). The three-month frame took into account the bank vole breeding period from gestation until weaning [26,58] and the response of the surrounding vegetation to vole abundance (see [59] for a similar approach). The 6-month frame accounted for the period between consecutive sampling sessions, and the 12-month frame was used to account for the life expectancy of the vole in the wild [26]. To address possible changes in the frequency and/or intensity of extreme weather events [60], the variance of meteorological variables was included [61]. These variables may reflect the ecological effects that climate variability could have, in the environment of a small mammal. Temperature may affect thermoregulation, while precipitation and drought conditions may affect primary productivity.

Land use data were obtained from the Department of Climate Action, Food and Rural Agenda of the Catalan government [62]. Maps from different years (2007, 2012, 2017, 2022) were available. A 500 m buffer was created around each sampling station and then land cover was extracted from the buffer. This buffer area was ideal for describing habitat features at the landscape level for small mammals, showing spatial homogeneity in land-use changes between time periods [39]. The land cover classifications (24) were reorganized into four main categories: forests, open spaces (i.e., grassland, scrubland, and post-fire habitats), urban areas, and crops [57]. Forests included sclerophyllous, deciduous, and coniferous forests and were maintained as sub-categories considering the habitat requirements of the species (see [59] for a similar approach). Land-use covers for the years without data (e.g., 2008–2011) were obtained by linear interpolation, and changes between periods were evaluated using linear regression models.

### 2.2. Data Analysis

#### 2.2.1. Abundance Models

The response variables were the total abundance (adults + subadults + indeterminate), the abundance of adults, and the abundance of subadults (the threshold between maturity was 18 g weight [63], also coinciding with the fact that, below this weight, no individual in our sample was reproductively active), only including different individuals (i.e., without recaptures). Predictors were habitat cover (percent cover: forest, open, urban and crops), sampling period (categorical: spring or autumn), meteorological variables (mean temperature, precipitation, and water deficit, as well as their variance) for the three selected periods (three, six, and twelve months prior to the survey) and the region (Pyrenees or Mediterranean). The first selection of the variables to be tested was made by a correlation matrix according to the strength of association (Spearman correlation: ρ ≥ 0.75 positive and negative) and their ecological significance for the species. Then, a multicollinearity test was performed to decide the final variables by the Variance Inflation Factor (VIF), selecting sets of independent variables with VIF values < 5 [64]. These left the following variables as final predictors in the statistical models: sampling period, mean temperature, cumulative precipitation, temperature variance, precipitation variance twelve months prior to the survey, percentage forest cover, and region (Pyrenees or Mediterranean). Interannual population synchrony was assessed by regressing the mean population abundance in a region (i.e., Pyrenees) with the mean population in the other region (Mediterranean) also considering a one-year lagged response. Seasonal population synchrony followed the same approach but considering seasonal (i.e., spring or autumn) mean population abundance as the response variable.

The general linear mixed effects models were executed with the lme4 package [65] to test the association between the response variable and the fixed factors (year, climate, land use, and region), and the interaction between the sampling period and the region was included. The three statistical models for bank vole abundance (total, adults, subadults) were built using a negative binomial error distribution instead of Poisson [65,66] as it was the best fit according to the goodfit function test of the vcd package [67], with explanatory variables scaled prior to model fitting. Random effects included the station, plot, and sampling session/campaign [68,69]. Five different models were built: total and adult abundance with weather variables 12 and 6 months prior to sampling, and subadult abundance 3 months prior to sampling. The *dredge* function was used to evaluate models with all possible combinations of explanatory variables [70]. Parsimony was evaluated using the corrected Akaike information criterion (AICc) [71], considering only significant models not exceeding ∆AICc < 2. It was decided to present the best model obtained, since interactions between variables had been proposed so it was not possible to perform a model averaging. The significance level for this study was set at 0.05; a case of *p* < 0.1 was considered as the trend. The data were processed under the R and R-Studio environments [72].

#### 2.2.2. Occupancy Models

Small mammals are secretive species whose presence could be overlooked unless detectability is explicitly integrated into occupancy models and local colonization and extinction can be measured without error. To determine the parameters that can influence the occupancy dynamics of the bank vole, multiple-season single-species occupancy models were used [18] calculated through the unmarked package [73]. In these models, it is considered that the occupancy in a place in the current season depends on the state of occupancy in that place in the previous season (first-order Markovian changes [18]). Furthermore, the models also take into account the imperfect detection of the species by calculating the detection probabilities [74]. Through these models, the probabilities of occupancy, local colonization, local extinction, and detection were obtained. Occupancy (ψ) is the proportion of sites that are occupied by the target species. Colonization (γ) is the probability that an unoccupied place at season *t* is occupied by the species at season *t* + 1. Extinction (ε) is the probability that a site occupied at season *t* will be vacated by the species at season *t* + 1. Detectability (*p*) is the probability of detecting the species when actually present [18].

Occupancy models tested total occupancy (considering adults, subadults, or indeterminates) and included the variables of habitat (annual interpolated land cover change between 2007 and 2022), vegetation structure of the understorey (vegetation cover < 1 m, through LiDAR data [75]), region (Pyrenees/Mediterranean), and season (spring/autumn). Again, a model selection with the *dredge* function was performed and, in this case, as no interactions between variables were considered, we selected the first model created by averaging using the MuMIn package [68].

#### 2.2.3. Analysis of Population Trends

Temporal trends in bank vole population abundance were assessed using TRIM (TRends and Indices for Monitoring data; Pannekoek and Strien, 2005), by means of *r-trim* R package version 4.4.0. [76]. This methodology analyses a time series of log-transformed count data, accommodating and imputing missing observations. TRIM Model 2 was used, employing a linear trend with a constant mean growth rate across sites and accounting for overdispersion. The number of individuals served as the response variable to calculate the abundance index, with plot identity and sampling campaign designated as site and time covariates, respectively. Missing observations were calculated as imputed counts by the model. Trends in bank vole occupancy were evaluated by building an occupancy model testing the interaction between the sampling campaign and region (Pyrenees versus Mediterranean) as a predictor while controlling for species detectability throughout the study period.

## 3. Results

### 3.1. Weather

The mean temperatures and cumulative rainfall at the beginning (2008) and end (2023) of the study period are presented in Table 1. Differences in temperature and rainfall between both regions were mostly associated with altitude, the Pyrenean stations being at higher mean elevation (X = 1594 ± 324 m versus X = 794 ± 429 m).

The temperatures increased and the precipitation decreased along the study period in both areas (Pyrenees and Mediterranean, all *p* < 0.001) (Appendix A). In the Pyrenees, there was a higher trend for the increase in temperatures (marginal) and a lower trend for the decrease in precipitation compared with the Mediterranean stations, but significant differences were not detected (Appendix A and Appendix A).

### 3.2. Land Cover Change (2007–2022)

The percentages of land-use coverage around the sampling plots by region, both at the beginning (2007) and at the end (2022) of the study period, are presented in Table 2.

In the Mediterranean area, there was an increase in urban and open habitats, and a decrease in forest and crops; however, the changes were non-significant. For the Pyrenees area, the trend showed an increase in urban and forest and a decrease in open and crops, but crops were the only marginally significant change (*p* < 0.1). Comparing the two areas, in the Pyrenees, there were more forests year by year (*p* < 0.05) and marginally less open areas (*p* < 0.1) compared to the Mediterranean sampling stations (Appendix A). The rest of the categories or subcategories did not show a significant difference (Appendix A).

### 3.3. Sampling Effort and Capture Success

With 16 trapping stations in the Mediterranean and 13 in the Pyrenees, the number of trapping sessions ranged from 4 to 32 per site, with 438 in total. The total sampling effort performed was 42,444 trap nights, yielding 7641 (18%) captures of small mammals. Captures included 16 species: Wood mouse (*Apodemus sylvaticus*, 44.3%), Bank vole (*Clethrionomys glareolus*, 19.7%), Greater white-toothed shrew (*Crocidura russula*, 11.9%), Yellow-necked mouse (*Apodemus flavicollis*, 7.89%), *Apodemus* sp. (6.39%), Algerian mouse (*Mus spretus*, 4.3%), Common shrew (*Sorex araneus*, 1.36%), Garden dormouse (*Eliomys quercinus*, 1.30%), Crowned shrew (*Sorex coronatus*, 0.98%), Edible dormouse (*Glis glis*, 0.73%), Pigmy shrew (*Sorex minutus*, 0.66%), European snow vole (*Chionomys nivalis*, 0.25%), Mediterranean field vole (*Microtus lavernedii*, 0.13%), Water shrew (*Neomys fodiens*, 0.07%), Common vole (*Microtus arvalis*, 0.04%), Mediterranean pine vole (*Microtus duodecimcostatus*, 0.01%), and House mouse (*Mus musculus*, 0.01%).

For the bank vole, the total number of captures was 1507 (including recaptures), identifying a total of 984 different individuals where 453 occurred in the Mediterranean and 531 in the Pyrenees, both with a reduced mortality rate (0.46% in total, 7 individuals). The mean number of individuals captured per sampling session was 3.83 ± 0.39 (SE). Regarding sex-ratio, the captures were 51.6% females and 44.9% males (the remaining 3.45% were undetermined). There was a higher capture rate of adults (83.9%) than of subadults (16.1% weighing less than 18 g). In contrast, most of the catches were of sexually inactive individuals (60.5%), followed by active (27.5%) and undetermined (11.9%). The years with the highest abundance coincided between regions (e.g., 2009, 2014, 2021, Figure 2), but there was a seasonal time lag, with the peak in the Mediterranean in the spring and the peak in the Pyrenees in the following autumn (*p* < 0.1). The models on population synchrony revealed moderate interannual synchrony in both regions (R^2^ = 0.28, n = 16, *p* < 0.05), and a delayed seasonal synchrony (R^2^ = 0.22, n = 30, *p* = 0.06), suggesting that the abundance in the Pyrenees was delayed by one season.

### 3.4. Abundance Models

For total and adult abundances, the 12-month models were selected as they provided a better fit than the 6-month models (Total abundance 12 Month: R^2^c = 0.53; R^2^m = 0.16 vs. 6 Month: R^2^c = 0.50; R^2^m = 0.10, Adult abundance 12 Month: R^2^c = 0.53; R^2^m = 0.14 vs. 6 Month: R^2^c = 0.49; R^2^m = 0.08). Additionally, the 6-month models failed to meet the variable selection criteria due to multicollinearity issues during the Variation Inflation Factor (VIF) test. As a result, only the 12-month and 3-month models are presented (Table 3, Figure 3).

For total abundance, the variables selected by the model (given by the dredge function) included the mean temperature (12 months before), precipitation variance (12 months before), the coniferous forest, and the interaction between season and region. The model suggested that higher temperatures and variance in precipitation during the year before the captures, and the cover of coniferous forests, decreased the abundance of bank voles, and, in the Pyrenees, the abundance was higher in autumn. For adults, the results were similar, with mean temperature, precipitation variance, and coniferous forest, all having a negative effect. Finally, subadults showed a significant response in their abundances to the interaction between autumn and Pyrenees and the variance of temperature three months prior sampling. In this case, the season–region interaction had a positive effect in the abundance, and appeared to also have a positive effect on the variance of the temperature, a variable that did not appear in the other models (Table 3, Figure 3) In other words, the Pyrenees in autumn are more likely to have a higher abundance of subadults which supports the results for the total abundance model and the mean abundances (Figure 2). *R-trim* analysis indicated contrasting population trends, with an increase in the Mediterranean (+116% in 16 years) and a decrease in the Pyrenean plots (−44%), though the region-by-time interaction was only marginally significant (Wald test = 2.97, *p* = 0.08, Figure 2b).

### 3.5. Occupancy Models

After performing the Dredge function, six models were selected as having a better fit than the null model. But, none of the models selected predictors that affected colonization (γ), detectability (*p*), or extinction (ε) (Appendix A). However, naïve and fitted occupancy (Ψ) increased along the study period in the Mediterranean (+16.5%) and decreased in the Pyrenees (−1.5%, Figure 4b, Appendix A). Although occupancy showed a similar pattern than the population trends in both regions, expansions and retractions of the species range were not confirmed (interaction campaign × region: z = 1.5, *p* = 0.13).

## 4. Discussion

To our knowledge, this is the first study that considers a long time series (16 years) to analyze the influence of habitat and climate change on bank vole populations and demography in its southernmost range in Western Europe.

Despite using a relatively short time series to detect climatic trends, the results are in general agreement with the knowledge on climate change in the study area and elsewhere in the Mediterranean region and the combined effects of increasing temperatures and decreasing precipitation [77]. Nonetheless, there was no evidence of relevant land cover changes at the plot level in the Mediterranean area, as previously described in other plots of the same study area [39], though they were confirmed in the Pyrenees (i.e., afforestation).

### 4.1. Population Dynamics of Bank Voles: The Effects of Climate and Habitat

The population dynamics of bank voles showed interannual and seasonal variations throughout the study period, with regional differences. The fixed factors (climate and land cover) included in the statistical models explained only a small portion (5–16%) of the variability in abundance, highlighting that other untested variables (e.g., food availability, density dependence, and predation) also played a role. Small mammals of the temperate zone reproduce seasonally as they need to match the birth period with the period when environmental conditions (i.e., food and climate) are optimal [49]. The abundance of bank voles increased from spring to autumn in the Pyrenees, suggesting that the species reproduced during the summer [58,78], the favorable season in that area. This contrasted with the dynamics in the Mediterranean area, with populations decreasing from spring to autumn, suggesting that summer is a limiting season for breeding in Mediterranean rodents [59]. Our results indicated that higher mean temperatures in the previous year were associated with lower abundance of bank voles, and subadult abundance was affected by the variability of mean temperatures in the previous three months, suggesting that reproduction and recruitment are particularly sensitive to recent climatic conditions. Other studies indicated even longer term negative effects of temperature (two years of previous sampling [31]), suggesting that there are indirect effects of temperature on bank voles via masting events. This likely reflects physiological and ecological constraints on the species, as higher temperatures may exacerbate heat stress, reduce food availability, or disrupt breeding cycles. Adaptive genes of bank voles were associated to annual mean temperatures, suggesting that southern populations can be better adapted to warmer temperatures and drier climates than northern populations [79]. While some studies have documented the influence of precipitation on abundance and activity [80,81], this study did not detect a direct effect. However, precipitation variance exhibited a negative association with bank vole abundance. This suggests that extreme weather events (e.g., floods, [82]) may be playing a role in bank vole demography by decreasing growth rates, as occurs in many animal species [83]. In fact, these results are consistent with responses observed in other Mediterranean small mammals that were negatively affected by short-term extreme weather events in the same area (*Crocidura russula* [75] and *Mus spretus* [57]).

Although there have been authors who have included ‘habitat’ in their analyses, such as type of cover (forest, shrubland, grasslands) [25,30,84], silviculture treatments [85,86], or understorey cover [80], to our knowledge, no studies have explicitly differentiated land-cover variations between forest types based on their composition (coniferous, deciduous, and sclerophyllous). In our study, forest composition played a pivotal role on bank vole abundance. Coniferous forest expansion negatively impacted total and adult abundances, while deciduous forests provided more favorable conditions. Coniferous forests generally provide less diversified vascular understories than broadleaved forests, potentially affecting habitat suitability for bank voles [87]. This distinction underscores the role of forest type in shaping habitat quality for small mammals. In transitional zones between biogeographic regions, deciduous forests are declining at the expense of coniferous forests [88] and these shifts in forest composition may compromise the viability of bank vole populations in the future. This finding partially contradicts previous suggestions that this species could benefit from the ongoing afforestation process in the Iberian Peninsula [37]. However, when the specific type of afforestation occurring in the region is considered, these results become more understandable.

Our results highlighted interannual population synchrony in both regions (Pyrenees and Mediterranean) with some favorable years (peaks in 2009, 2014, 2021) throughout the whole study area. Three out of four years of population peaks occurring in the study period (16 years) coincided between regions. Rather interestingly, the four peaks observed in the Pyrenees (2009, 2014, 2017, 2021) roughly matched with the peaks noticed in Southern Norway [89]. Bank voles are known to synchronize populations over large geographical areas [90], but these results suggest population synchrony at higher spatial scales than previously described. In the Mediterranean region, bank voles showed a one-year delayed response to seed availability (beech and acorn, [88]), but bank voles were unaffected by these seeds in northern populations [89]. However, bank voles showed a one year delay with respect to bilberry seed production in Norway [89]. The interaction between region and season revealed a higher abundance in the Pyrenees during autumn. This could indicate a combination of favorable climatic conditions, such as cooler temperatures. It is likely that the Pyrenees serve as a climatic refuge for the species, buffering the more extreme conditions (for this species) of the Mediterranean region. Although peripheral populations are often considered more vulnerable than central populations [79], our findings indicate the opposite trend. Specifically, we observed a positive population trend in the Mediterranean and a negative population trend in the Pyrenees, albeit with marginal statistical significance. These patterns need to be confirmed in the future by including more years and sampling stations in our time series.

### 4.2. Occupancy Dynamics of Bank Voles

The occupancy models are especially suited for secretive animals such as small mammals, and these kinds of analyses are needed to deal with detectability issues. Low detectability will yield biased estimates of occupancy; however, in this case, bank vole detection probabilities (*p* = 0.70 ± 0.2) were well above the threshold of 0.3 [90], and similar to values observed in Andorra (*p* > 0.8, [46]), which means that there was a low chance of not detecting the species when present [90]. Despite the effects of climate and habitat on abundance, the occupancy models showed rather stable occupancy patterns in both regions across the years, suggesting that either ongoing climate or land cover change did not affect occupancy dynamics (see [56,57] for similar results). These results can be interpreted in the light of the small changes observed in land cover at the sampling stations, especially in the Mediterranean region (<1% forest change). This indicated that moderate landscape change (<5%) did not have a significant impact on the probability of occupancy, as was previously stated in other small mammals (e.g., *Mus spretus* [57]). Despite marginal regional significance in population trend analyses, congruent patterns of occupancy change indicated potential future range shifts. Specifically, under projected global change scenarios (i.e., increasing temperatures and afforestation), range expansions are anticipated in Mediterranean regions, while range retractions are expected in Pyrenean regions. The interactive population trends by region, together with the presumed bank vole adaptability under warmer conditions [79], provides evidence that potential climate-driven range retraction may not be as severe as previously estimated [32].

### 4.3. Future Research

Although climate and habitat variables were examined, other factors such as habitat fragmentation, connectivity, population cyclicity, predators, food availability, or density-dependence, were not directly assessed in this study. There is evidence that bank voles showed long-term population oscillations in northern latitudes [89,91], but we did not consider it because these cycles were not described in southern Europe (e.g., France and Italy [92,93]). Bank voles are associated with masting events in the study area [94] and elsewhere [95,96], and this could explain the low variance explained by the climatic variables in the statistical models [95,96]. However, fruit availability may be inferred from the climatic data, since masting events depend on the weather conditions of previous years [31,88,96]. Intrinsic regulation is common in rodent populations, and density-dependence in the breeding season is most likely due to emergent territorial behavior of females and competition for space in bank voles [97]. Considering all these factors together would allow for greater precision in modeling species abundance and occupancy [81,84,98,99].

## 5. Conclusions

This study sheds light on the complex interactions between habitat, climate, and population dynamics in the bank vole (*Clethrionomys glareolus*) at the southern edge of its distribution range. Our research highlights the importance of considering both habitat and climate variability when assessing the species’ abundance and distribution, particularly in regions where environmental conditions are changing.

The results underscore the sensitivity of bank voles to specific habitat types and climatic factors. Coniferous forests appear less favorable for the species, while variability in precipitation, especially during extreme weather events, has emerged as a significant driver of population fluctuations. The findings also suggest that warmer conditions in the preceding year may negatively impact bank vole populations, pointing to the broader implications of climate change for the species. Given these findings, forest management practices should prioritize the support of a mix of tree species and structural complexity since it could mitigate the negative impacts of climate change and coniferous forest expansion on bank vole populations. However, some authors suggested that bank voles can maintain climate adaptation under warming predicted for the next few decades [79], which can contradict the potential retractions expected based on climate niche models alone [32]. Additionally, future research should aim to integrate data on food availability, predator pressures, forest structure and composition, and genetic adaptability to better understand the factors influencing bank vole populations and to refine conservation strategies in peripheral areas at the southernmost range. By examining both abundance and occupancy [100], we can gain a more comprehensive understanding of the population dynamics of bank voles. Analysis of long-term data allowed us to fill critical gaps and improve our understanding of how small mammals respond to environmental changes in climatically and ecologically sensitive regions such as the Mediterranean.

## Figures and Tables

**Figure 1 animals-15-00839-f001:**
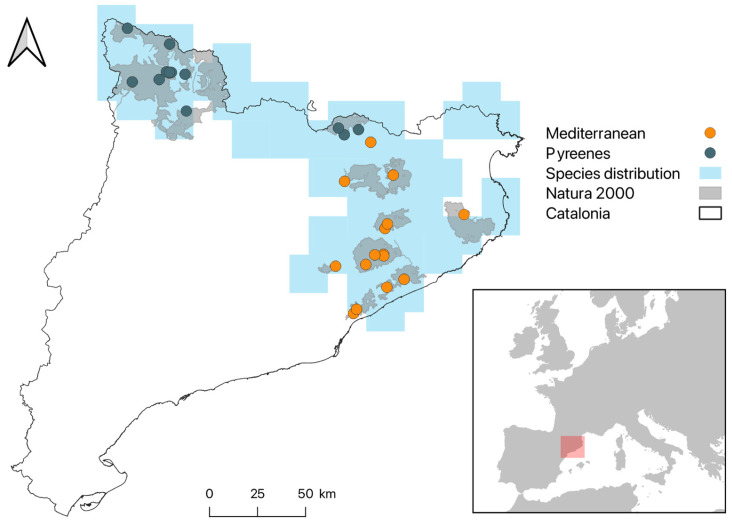
The distribution of the 29 SEMICE sampling stations where *Clethrionomys glareolus* was found in Catalonia, Spain. The light blue area is a zone of potential presence calculated from a 5 km buffer around the 10 × 10 squares in which the bank vole was known to occur based on the Atlas and Red Book of the Mammals of Spain [26]. The red square situates Catalonia regarding Europe.

**Figure 2 animals-15-00839-f002:**
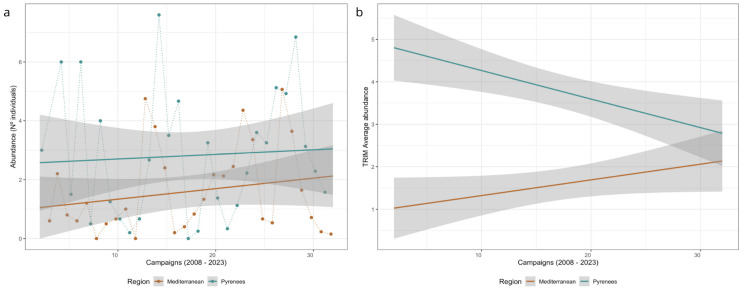
(**a**) Observed mean bank vole abundance per sampling campaign and region, and population trends along the study period, and (**b**) linear population trends by region modelled with *r-trim* including the imputed counts calculated from the missing values in the time series, taken account of overdispersion and series correlation (interaction sampling campaign × region: Wald test = 2.97, *p* = 0.08). Shaded areas are CI at 95%.

**Figure 3 animals-15-00839-f003:**
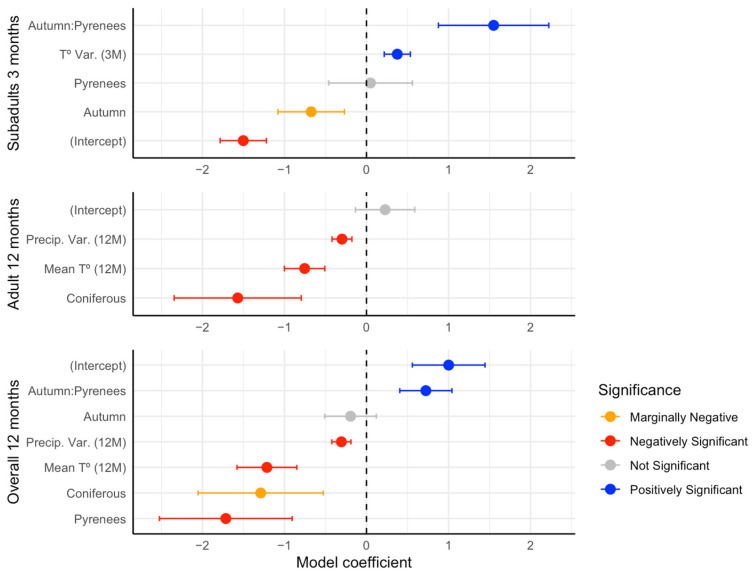
Results of the GLMMs for the subadults (**upper**), adult (**middle**), and total (**lower**) abundance of *Clethrionomys glareolus* as response variables, and a set of scaled climatic and land-use predictors. Coefficients other that zero are marked in red (negative effect) or blue (positive effect).

**Figure 4 animals-15-00839-f004:**
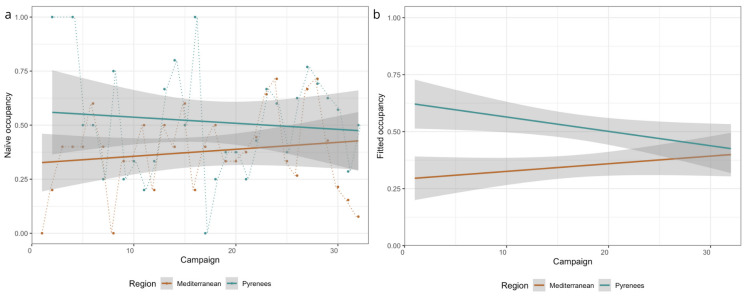
(**a**) Observed (naïve) and (**b**) fitted occupancy per sampling campaign and region (Mediterranean and Pyrenees), and occupancy trends along the study period (interaction sampling campaign × region: z = 1.5, *p* = 0.13). Shaded areas are CI at 95%.

**Table 1 animals-15-00839-t001:** Average temperature and accumulated precipitation by region in 2008 and 2023.

	Pyrenees		Mediterranean	
	2008	2023	2008	2023
Temperature	5.52° ± 0.45 °C	7.61° ± 0.49 °C	11.75° ± 0.40 °C	13.22° ± 0.46 °C
Precipitation	1652.57 ± 36.19 mm	921.45 ± 32.93 mm	1248.58 ± 59.85 mm	510.63 ± 28.84 mm

**Table 2 animals-15-00839-t002:** Percentage (%) of land cover at the sampling plots by region in 2007 and 2022.

	Pyrenees		Mediterranean	
	2007	2022	2007	2022
Forest	77.9 ± 5.29	84.7 ± 4.24	85.8 ± 4.24	85.0 ± 4.32
Open	19.9 ± 4.54	15 ± 4.23	9.35 ± 4.09	10.2 ± 3.95
Crops	0.30 ± 0.19	0	3.25 ± 1.29	2.85 ± 1.07
Urban	0.29 ± 0.20	0.35 ± 0.20	1.48 ± 0.61	1.86 ± 0.66

**Table 3 animals-15-00839-t003:** Final general linear mixed model selected using the dredge function to evaluate the effect of the percentage of forest cover and climatic variables on the total, adults, and subadults abundance. Station, plot, and sampling session were included as random effects. The first value represents the estimate, the value in brackets represents the standard error, R^2^m represents the marginal R-squared, a statistical measure that refers to the variance explained by fixed factors, and R^2^c represents the conditional R-squared. The variance is explained by both fixed and random factors.

	Total Abundance	Adults	Subadults
Intercept	1.00 **	227	−1.50 ***
	(0.44)	(0.36)	(0.28)
Coniferous forest	−1.28 *	−1.57 **	
	(0.76)	(0.77)	
Autumn	−0.19		−0.67 *
	(0.31)		(0.40)
Pyrenees region	−1.71 **		0.04
	(0.80)		(0.51)
Precipitation Variance	−0.30 ***	−0.29 **	
(12 month before)	(0.11)	(0.12)	
Mean temperature	−1.21 ***	−0.75 ***	
(12 month before)	(0.36)	(0.24)	
Temperature Variance			0.37 **
(3 month before)			(0.15)
Autumn:Pyrenees region	0.72 **		1.55 **
	(0.31)		(0.67)
R^2^m	0.16	0.14	0.05
R^2^c	0.53	0.53	0.05

Significance values: 0 = ‘***’, 0.001 = ‘**’, and 0.01 = ‘*’.

## Data Availability

The data presented in this study are available on request from the corresponding author.

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
