# Peer review of "Effects of Climate and Land Use on the Population Dynamics of the Bank Vole (Clethrionomys glareolus) in the Southernmost Part of Its Range"

_animals, 2025, doi:10.3390/ani15060839_

Round 1

Reviewer 1 Report

Comments and Suggestions for Authors

Manuscript ID: animals-3360228

Effects of climate and land use on the population dynamics of the bank vole (Clethrionomys glareolus) in the southernmost part of its range

By: Lucía de la Huerta - Schliemann *, Marc Vilella, Lídia Freixas, Ignasi Torre

Review

Manuscript concerns very important topic – climate change effects on ecosystems and animals. Authors analyze the impact of climate change and land use on the population dynamics of the bank vole at the southernmost edge of its range. The research is based on 16 years of data from the SEMICE (small mammal monitoring) program. Authors analyzed synergy of the  climate change, habitat changes, and small mammal population dynamics.

Overall, the study is well-done, methodologically is sound, and provides a comprehensive analysis. The results have some implications for conservation practices and forest management in Mediterranean and Pyrenean ecosystems. After revising, it could be published in Animals.

Some important findings are:

·       Highlighting the importance of considering both habitat and climate variability when assessing the species' abundance and distribution, particularly in regions where environmental conditions are changing

·       Absence of population fluctuation in the southern part of range of C. glareolus

·       Identification of the Pyrenees as a potential climate refuge for the species.

General comments

1.       I like the scope, as analyzed species, bank vole, is on the edge of its range in the Pyrenees. So far, absolute most of the analyses cover the northern part of the range in this species.

2.       Authors use clear writing style, explain used methodology, and interpret their results. They strengthen their text with additional files on the land use and climate parameters, but include these in Supplements.

However:

1.       C. glareolus is a northern species with adaptations to cooler climates; this must be mentioned, in Simple summary, Abstract and Introduction. In Spain, species is on the southern end of its range, therefore, inhabits specific ecosystems;

2.       While authors cite some sources referring to bank voles in high latitudes, I think they might consider lower latitudes also, e.g., citing Diversity 2024, 16(9), 546; https://doi.org/10.3390/d16090546  and Nat Commun 14, 7840 (2023); https://doi.org/10.1038/s41467-023-43383-z. Of course they are free to choose the other ones.

3.       As alternate factors, genetic diversity should be considered as it was mentioned is several sources, e.g.,  Commun Biol 7, 863 (2024);  https://doi.org/10.1038/s42003-024-06549-z

4.       Authors might consider to integrate supplementary materials into the main manuscript text.

5.       As for 18 g body weight, used as a threshold for adult animals. This might be the most critical point in your paper. The cited reference is from the north… please elaborate, using the range of countries where snap-trapping data are available, therefore adult body mass was evaluated under dissection. A group of subadult animals is missing on your study.

6.       Line 275: consider using (X = 1594 ± 324 m versus X = 794 ± 429 m).

7.       Back matter: expand Conflicts of Interest about the possible role of funders, use journal Template to find recommended text.

Specific Comments

1.       Line 122: ”changes” missing?

2.       Lines 145-146: is December an autumn month, and is June a spring month? If yes, please give some description on the seasonality in Spain, as an international readers will be confused;

3.       Line 169: use “200–2066 m”, i.e., “–“ for all ranges;

4.       Line 195: please separate m from numbers, as was cm in Line 148; check rest of the text;

5.       Check for mistypes, e.g., Line 639;

6.       Lines241-242: as later you use p <0.1, please consider introduce this level here, and call it “the trend”;

7.       The finding of stable occupancy is intriguing but possibly conflicts with the abundance patterns (see Figure 2). This discrepancy could be explored further, possibly by linking to metapopulation dynamics.

8.       In the Figure 4, years and seasons are not shown – but they are mentioned in the text, Line 356; also, there must be differences between minimum and maximum occupancy values, as this seems visually. What are whiskers – SE or SD? These thing should be in the caption, or in Methods;

9.       As already mention, expand references to include countries of middle latitudes.

Figures and Tables

1.       Figure 2: please explain, why _A  and _S are used with the year;

2.       Figure 3: is there a red color? Please check, if caption is clear.

3.       Figure 4: expand caption for figure to be clear explained

4.       Supplementary tables and figures are referred in the main text (as Table S1, Figure S1, etc.), however, please consider transferring Supplementary Figures to the main text.

Conclusion

The manuscript suits to SI it is targeting, particularly with the context of edge populations and their adaptation to climate and habitat changes. With revisions to expand context, refine figures, and address alternative factors, this study has the potential to be published in Animals.

Author Response

C. glareolus is a northern species with adaptations to cooler climates; this must be mentioned, in Simple summary, Abstract and Introduction. In Spain, species is on the southern end of its range, therefore, inhabits specific ecosystems

Answer 1: Thank you for pointing this out, it has been included in all three sections.

While authors cite some sources referring to bank voles in high latitudes, I think they might consider lower latitudes also, e.g., citing Diversity 2024, 16(9), 546; https://doi.org/10.3390/d16090546  and Nat Commun 14, 7840 (2023); https://doi.org/10.1038/s41467-023-43383-z. Of course they are free to choose the other ones.

Answer 2: Thank you for your input, it has been a great help, they have been considered and two of them have been included in the references used throughout the article. 

As alternate factors, genetic diversity should be considered as it was mentioned is several sources, e.g.,  Commun Biol 7, 863 (2024);  https://doi.org/10.1038/s42003-024-06549-z

Answer 3: Thanks for the suggestion, but we think that the citation in Nature Communications is more suitable for the issues regarding genetic adaptability to warming.

Authors might consider to integrate supplementary materials into the main manuscript text.

Answer 4: Thank you for the comment, we have included some of the elements of the supplementary material in the body of the article, following your recommendation. 

As for 18 g body weight, used as a threshold for adult animals. This might be the most critical point in your paper. The cited reference is from the north… please elaborate, using the range of countries where snap-trapping data are available, therefore adult body mass was evaluated under dissection. A group of subadult animals is missing on your study.

Answer 5: The expression and grouping of individuals have been modified so that juveniles and subadults are considered within the group ‘subadults’. Despite this weight threshold (set at 18g) not being appropriate to our study area, we have seen that all the individuals weighing 18g or less were non-breeding, so this will help us to use this threshold to separate subadults from adults. We included this in the methods section to give support to the age/weight criterion. The use of weight from dead individuals (snap-trapping) maybe cannot be directly comparable to weight from live individuals. The time elapsed before weighing can influence the weight, as decomposition or dehydration can alter the mass.

Line 275: consider using (X = 1594 ± 324 m versus X = 794 ± 429 m).

Answer 6: Thank you for pointing this out, it has been considered and modified

Back matter: expand Conflicts of Interest about the possible role of funders, use journal Template to find recommended text.

Answer 7: It has been modified, thanks for pointing this out.

Specific Comments

Line 122: ”changes” missing?

Answer 1: Thank you for pointing this out, it has been modified

Lines 145-146: is December an autumn month, and is June a spring month? If yes, please give some description on the seasonality in Spain, as an international readers will be confused;

Answer 2: We used the astronomical calendar, which is not correct. Thank you for pointing this out, it has been modified. Our Spring (March to May) and Autumn (September to November) groups included the beginning of summer (June) and winter (December), respectively, in each campaign. This has been specified and included in materials and methods.

Line 169: use “200–2066 m”, i.e., “–“ for all ranges;

Answer 3: Thank you for pointing this out, it has been modified.

Line 195: please separate m from numbers, as was cm in Line 148; check rest of the text;

Answer. 4: Thank you for pointing this out, it has been modified.

Check for mistypes, e.g., Line 639;

Answer 5: Thank you for pointing this out, it has been modified.

Lines241-242: as later you use p <0.1, please consider introduce this level here, and call it “the trend”;

Answer 6: Thank you for pointing this out, it has been added.

The finding of stable occupancy is intriguing but possibly conflicts with the abundance patterns (see Figure 2). This discrepancy could be explored further, possibly by linking to metapopulation dynamics.

Answer 7: We acknowledge the discrepancy and conducted additional analyses to clarify this issue. Generalized linear mixed models (GLMMs) of bank vole abundance did not directly assess population trends (i.e., temporal changes in abundance) because the sampling campaign (1-32) was treated as a random factor, precluding its inclusion as a fixed factor for trend analysis. Consequently, we employed R-trim, a method specifically designed for time-series analysis of log-transformed count data. R-trim accommodates missing data, accounts for overdispersion and serial correlation, and we consider it more appropriate for examining population trends, particularly when incorporating a site covariate to differentiate regional trends. This analysis revealed a marginally significant regional difference in population trends (p = 0.08), indicating an increase in abundance in the Mediterranean and a decrease in the Pyrenees. Furthermore, a separate simple model examining the interaction between sampling campaign and region for occupancy was conducted, conceptually mirroring the R-trim analysis. This model also demonstrated a marginal regional trend in occupancy (p = 0.13), consistent with the population trend results. However, it is important to note that these results remain suggestive but statistically non-significant.

In the Figure 4, years and seasons are not shown – but they are mentioned in the text, Line 356; also, there must be differences between minimum and maximum occupancy values, as this seems visually. What are whiskers – SE or SD? These thing should be in the caption, or in Methods;

Answer 8: Figures 2 and 4, depicting population and occupancy trends, respectively, have been revised for direct comparability. Specifically, Figures 2-4a now illustrate the observed population and occupancy trends by region, while Figures 2-4b present the fitted statistical models. To ensure consistent representation and facilitate direct comparison, the dispersion of mean abundance values has been omitted, as corresponding dispersion statistics are not applicable to naïve occupancy.

As already mention, expand references to include countries of middle latitudes.

Answer 9: It has been considered and added, thanks for highlighting this point.

Figures and Tables

Figure 2: please explain, why _A  and _S are used with the year;

Answer 1: Thank you for pointing this out, it has been modified in the description of the figure.

Figure 3: is there a red color? Please check, if caption is clear.

Answer 2: Thank you for pointing this out, colors have been modified for a better representation of the graph.

Figure 4: expand caption for figure to be clear explained

Answer 3: We added information to make the caption of the figure more clear.

Supplementary tables and figures are referred in the main text (as Table S1, Figure S1, etc.), however, please consider transferring Supplementary Figures to the main text.

Answer 4: Thank you for pointing this out, it has been considered and some of the tables have been moved to the main text.

Conclusion

The manuscript suits to SI it is targeting, particularly with the context of edge populations and their adaptation to climate and habitat changes. With revisions to expand context, refine figures, and address alternative factors, this study has the potential to be published in Animals.

General answer: Thanks for the time and comments in order to improve our work, they have been of great help to enhance the quality of our article. 

Reviewer 2 Report

Comments and Suggestions for Authors

Dear Authors,

I have read your manuscript „Effects of climate and land use on the population dynamics of the bank vole (Clethrionomys glareolus) in the southernmost part of its range” and I find it of interest. The study aims to evaluate how climate and habitat affected several population parameters such as the abundance, age classes and occupancy of the bank vole in two regions situated at the southern limit of its distribution in Western Europe. The study reveals the negative effect of climate changes (increased temperatures and reduced rainfall) and changes in land use, especially the forest structure (coniferous versus deciduous) on these border bank vole populations. The research has also practical implications, highlighting the need to maintain or even enhance diversity of forests to ensure preservation of wildlife in the conditions of the changing climate.

Overall the manuscript is well written and easy to read, but there are a few paragraphs that lack the clarity. I recommend the manuscript for publication, but I have a few concerns that should be addressed beforehand.

You speak of land use change but there is not only the change along the research period but also the variation among study sites, so it is not correct to interpret the entire effect of land use in terms of change. I think you need to discuss this. In addition, in the methods section you mention that habitat change was one of the predictors, but actually you have the values of land use covers in each site/year, not the measures of change (at least that is what one can understand from the text).

Concerning the age structure, you interpret the abundance of juveniles as a measure of reproduction intensity, but this is not exactly true. If there are few adults you cannot have many juveniles even if they have reproduced intensely. Therefore, you should rather (or also) consider the ratio of juveniles to adults as a measure of reproduction intensity.

In the conclusion section you mention the negative impacts of forest expansion on bank vole populations. As the bank vole is a forest species, this appears to be counterintuitive, so you should develop on this topic, explaining the reasons for its sensitivity to the forest type (composition) and the caracteristics of the reforestation process. Is the structure of the natural forests preserved during the reforestation or is it a change in composition (conifer forests taking place of former deciduous forests or vice versa)?

Concerning the past weather conditions, it is not very clear to me. Is it the monthly mean temperature/cummulative precipitations 3/6/12 months prior to the survey or is it the mean temperature/ cummulative precipitations of the last 3/6/12 months?

In the methods sections you say: „The null model was calculated for each response variable and compared with the best model obtained by the dredge through an ANOVA”, but these results are not to be found in the manuscript, neither in the main text, nor in the supplemetary material. In addition, ANOVA (i.e., the F test) is not appropriate for comparison of GLMMs, the F test assuming the normal distribution. You should use the likelihood ratio test instead, although it is not customary to mix model selection based on information criteria (AIC here) with model comparison based on statistical tests.

For the best models „the overdispersion and chi-square were also calculated” – for the negative binomial GLM(M)s, overdispersion is not an issue, as negative binomial distribution has a parameter accounting for it, so I do not see the point in this. The chi-square of the model? Is there even such a thing? The chi-square test statistic is calculated during model comparison, not for a particular model, as far as I know.

Other minor comments:

Line 96 – „Similary to” instead of „As”

Lines 101-102 – „its” instead of „their” – you speak of the bank vole as a species

Line 167 - there is one trapping grid per station or several? All 29 stations were surveyed in each trapping sessions? If not, the total sample size (in terms of stations/trapping sessions) should be reported.

Line 214 – you need to clarify how you used the correlation analysis to do the first selection of the variables. It is not clear.

Line 230 – you chose negative binomial distribution over Poisson because ou had overdispersion?

Line 242 – „assuming a probability of alpha error of 5%” – may be deleted, it is redundant with the significance level of 0.05.

Line 246 – „the probability of ‘false absence’” - this is not detectability, it is only related to it.

Lines 262-263 – somewhere you forgot a )

Lines 265-266 – „...was selected by averaging from the MuMIn package” you mean „...was created by averaging using functions from the MuMIn package”?

Line 275 – „... being Pyrenean stations at...” should be „... the Pyrenean stations being at…”

Line 278 – 0.001

Line 281 – “non-detected” - not detected? refering to the lack of significance for the interaction between region and year?

Line 293 – “barely significant” – marginally significant?

Lines 320-321 – are these r or r-squared? Since it is linear regression, it should be r-squared.

Line 322 – “delayed on a seasonal basis” – you mean “delayed one season”?

Lines 327-328 – you should mention somewhere what R2c (and R2m in the supplementary material) stands for. And because you are interested in the effects of weather and land use, it would be more correct to report (also) R2m.

Line 330 – “ only the 12-month models are presented here” – but in Table S5 there are also the models for 6 months.

Line 335 – “… had a negative effect in the abundance, except for the interaction” - this is not relevant as the two predictors are categorical and you do not mention which are the reference levels.

Line 340 – “the same” - actually it is not exactly the same

Line 340 – “than in the total abundance model” – it is “as in…”, but it is suoerfluous, so I suggest deleting.

Line 340-341 – “juveniles showed significance in their abundances in the interaction of autumn and Pyrenees” - “juveniles showed significant response of their abundances to the interaction between season and autumn” – the interaction is between (or among) predictors, the coeficient is for their levels.

Line 346 – Figure 3?

Figure 3 - I do not see the point in representing the intercept.

Line 355 - in table S6 extinction appears to vary between regions and with the change in forest cover.

Line 355 - what does static mean here?

Figure 4 - What does the blue line represent? If it is a linear regression, results should be presented. If the regression is not significant (if that is what static means), there is no reason for it to be on the graph.

Line 375 – other variables not only may, but definitely play a role in abundance variation.

Line 441 – “habitat variables were thoroughly examined” – I think there are many other habitat variables that may be considered - such as habitat fragmentation and connectivity or vegetation characteristics.

Line 468 – “more detailed data on food availability, predator pressures” – actually this study does not include data on food availability or predator pressures at all.

Line 474 – “climatically marginal” – what does this mean?

Supplementary Material

Table S1, S2 – Model instead of Modelo

Table S1-S4 – t instead of T

Table S3 - As there is no significant result here, I do not see the point in this table. One note in the caption of Fig. S2 would be enough.

Table S5 – Aren’t these GLMMs? You should give more details in the caption on the response variable(s) and random factors, as well as the significance of R2m and R2c and of the values in the parantheses. „Model selection using the Dredge function first GLM model suggested” – this is not very clear. Please rephrase. What are the „*/////////” in the row for Log Likelihood? Akaike Inf, Crit, and Bayesian Inf, Crit, it’s . instead of ,  but best writein full.

Table S6 – The title is very cryptic. What models? You should explain what the values in the parantheses mean and what Int stands for. Do not let the reader guess... It is Delta AIC not just Delta. The values for Weight are all 1 in this table. As these are the best competing models (or at least that is what I understand), this is not correct. Weights need to add up to 1.

Comments on the Quality of English Language

Overall English language is good, but there are a few sentences that lack the clarity and need rephrasing. Some of them are mentioned in the comments section.

Author Response

Comment 1: You speak of land use change but there is not only the change along the research period but also the variation among study sites, so it is not correct to interpret the entire effect of land use in terms of change. I think you need to discuss this. In addition, in the methods section you mention that habitat change was one of the predictors, but actually you have the values of land use covers in each site/year, not the measures of change (at least that is what one can understand from the text).

Answer 1: It has been changed to ‘land cover’, you are right to point that out, and indeed, for abundance models what we used is a percentage of cover which was calculated annually as interpolation from observed values between the quinquennial land cover inventories (2007, 2012, 2017, 2022). For the occupancy models the variable is effectively the whole value of change along the study period (2007-2022). Also, in the abundance models we considered season as a random factor, this allowed us to eliminate the issue of variation between study sites that you mention.

Comment 2: Concerning the age structure, you interpret the abundance of juveniles as a measure of reproduction intensity, but this is not exactly true. If there are few adults you cannot have many juveniles even if they have reproduced intensely. Therefore, you should rather (or also) consider the ratio of juveniles to adults as a measure of reproduction intensity.

Answer 2: Population demography was analyzed using two age classes, defined by body mass: adults (>18g) and subadults (=<18g). The subadult class encompassed juveniles, including recently weaned individuals. This classification aligned with reproductive maturity, as no subadults exhibited breeding condition. Consequently, while the subadult class included recent recruitment, it also reflected other demographic processes affecting this cohort over extended periods. Therefore, changes in the subadult class cannot be solely attributed to reproductive intensity. We pointed out this in the text.

Comment 3: In the conclusion section you mention the negative impacts of forest expansion on bank vole populations. As the bank vole is a forest species, this appears to be counterintuitive, so you should develop on this topic, explaining the reasons for its sensitivity to the forest type (composition) and the caracteristics of the reforestation process. Is the structure of the natural forests preserved during the reforestation or is it a change in composition (conifer forests taking place of former deciduous forests or vice versa)?

Answer 3: We agree with that comment; this was a counterintuitive pattern owing the better performance expected for a forest dwelling species on an expanding forest environment. We speculated that this could be partially related to changes in forest composition associated to the process of landscape change. Particularly, in the study area, the process of landscape change produced the progressive substitution of deciduous forests by coniferous forests (Oro et al. 2024), which were more unsuitable for the species. It has been remarked in the last sentence of the conclusion

Comment 4: Concerning the past weather conditions, it is not very clear to me. Is it the monthly mean temperature/cummulative precipitations 3/6/12 months prior to the survey or is it the mean temperature/ cummulative precipitations of the last 3/6/12 months?

Answer 4: The weather variables are prior to the survey, it has been clarified in the text. 

Comment 5: In the methods sections you say: „The null model was calculated for each response variable and compared with the best model obtained by the dredge through an ANOVA”, but these results are not to be found in the manuscript, neither in the main text, nor in the supplemetary material. In addition, ANOVA (i.e., the F test) is not appropriate for comparison of GLMMs, the F test assuming the normal distribution. You should use the likelihood ratio test instead, although it is not customary to mix model selection based on information criteria (AIC here) with model comparison based on statistical tests.

Answer 5: We initially included that as a verification step, however, as you pointed out, it was not statistically valid. We appreciate you bringing this to our attention and have removed it from the article.

Comment 6: For the best models „the overdispersion and chi-square were also calculated” – for the negative binomial GLM(M)s, overdispersion is not an issue, as negative binomial distribution has a parameter accounting for it, so I do not see the point in this. The chi-square of the model? Is there even such a thing? The chi-square test statistic is calculated during model comparison, not for a particular model, as far as I know.

Answer 6: As in the previous section, we used that as a double-check, but it seems our explanation wasn't clear. To prevent misunderstandings, we've removed it from the article. Thank you for pointing this out.

Other minor comments:

Comment 7: Line 96 – „Similary to” instead of „As”

Answer 7: Thank you for pointing this out, It has been modified

Comment 8: Lines 101-102 – „its” instead of „their” – you speak of the bank vole as a species

Answer 8: Thank you for pointing this out, it has been modified

Comment 9: Line 167 - there is one trapping grid per station or several? All 29 stations were surveyed in each trapping sessions? If not, the total sample size (in terms of stations/trapping sessions) should be reported.

Answer 9: Thank you for pointing this out, there is one trapping grip per sampling station, and not all 29 stations were always surveyed, it has been clarified in the text.

Comment 10: Line 214 – you need to clarify how you used the correlation analysis to do the first selection of the variables. It is not clear.

Answer 10: Thank you for pointing this out, correlations were done through a spearman correlation matrix, this has been clarified in the text.

Comment 11: Line 230 – you chose negative binomial distribution over Poisson because ou had overdispersion?

Answer 11: We use a negative binomial error distribution instead of Poisson as it was the best fit according to the goodfit function test of the vcd package of R, it has been clarified in the text. 

Comment 12: Line 242 – „assuming a probability of alpha error of 5%” – may be deleted, it is redundant with the significance level of 0.05.

Answer 12: Thank you for pointing this out, it has been removed.

Comment 13: Line 246 – „the probability of ‘false absence’” - this is not detectability, it is only related to it.

Answer 13: Thank you for pointing this out, it has been removed.

Comment 14: Lines 262-263 – somewhere you forgot a )

Answer 14: Thank you for pointing this out, it has been modified.

Comment 15: Lines 265-266 – „...was selected by averaging from the MuMIn package” you mean „...was created by averaging using functions from the MuMIn package”?

Answer 15: Thank you for pointing this out, it has been modified.

Comment 16: Line 275 – „... being Pyrenean stations at...” should be „... the Pyrenean stations being at…”

Answer 16: Thank you for pointing this out, it has been modified.

Comment 17: Line 278 – 0.001

Answer 17: Thank you for pointing this out, it has been modified.

Comment 18: Line 281 – “non-detected” - not detected? refering to the lack of significance for the interaction between region and year?

Answer 18: Thank you for pointing this out, it has been modified.

Comment 19: Line 293 – “barely significant” – marginally significant?

Answer 19: Thank you for pointing this out, it has been modified.

Comment 20: Lines 320-321 – are these r or r-squared? Since it is linear regression, it should be r-squared.

Answer 20: Thank you for pointing this out, it has been modified to the r-squared value.

Comment 21: Line 322 – “delayed on a seasonal basis” – you mean “delayed one season”?

Answer 21: Thank you for pointing this out, yes! It has been modified.

Comment 22: Lines 327-328 – you should mention somewhere what R2c (and R2m in the supplementary material) stands for. And because you are interested in the effects of weather and land use, it would be more correct to report (also) R2m.

Answer 22: Thank you for pointing this out, it has been modified and included the R2m in the text.

Comment 23: Line 330 – “ only the 12-month models are presented here” – but in Table S5 there are also the models for 6 months.

Answer 23: Thank you for pointing this out, it has been modified and 6 month models have been removed.

Comment 24: Line 335 – “… had a negative effect in the abundance, except for the interaction” - this is not relevant as the two predictors are categorical and you do not mention which are the reference levels.

Answer 24: Thank you for pointing this out, it has been modified.

Comment 25: Line 340 – “the same” - actually it is not exactly the same

Answer 25: Thank you for pointing this out, it has been modified.

Comment 26: Line 340 – “than in the total abundance model” – it is “as in…”, but it is suoerfluous, so I suggest deleting.

Answer 26: Thank you for pointing this out, it has been deleted

Comment 27: Line 340-341 – “juveniles showed significance in their abundances in the interaction of autumn and Pyrenees” - “juveniles showed significant response of their abundances to the interaction between season and autumn” – the interaction is between (or among) predictors, the coeficient is for their levels.

Answer 27: Thank you for pointing this out, it has been modified.

Comment 28: Line 346 – Figure 3?

Answer 28: We refer to Figure 2 in that sentence.

Comment 29: Figure 3 - I do not see the point in representing the intercept.

Answer 29: Thank you for pointing this out, in this case we prefer to keep it s.

Comment 30: Line 355 - in table S6 extinction appears to vary between regions and with the change in forest cover.

Answer 30: It is true, this variable has been selected by the dredge in the third and subsequent models. However, none of the models showed a significant difference, that’s why they are not discussed here. 

Comment 31: Line 355 - what does static mean here?

Answer 31: Thank you for pointing this out, actually is a mistake, so it has been modified

Comment 32: Figure 4 - What does the blue line represent? If it is a linear regression, results should be presented. If the regression is not significant (if that is what static means), there is no reason for it to be on the graph.

Answer 32: You are right, and this figure was nonsense owing to that we expected that occupancy patterns to be different between regions along the study period. Therefore, we changed this figure showing the observed (naïve) occupancy by region and sampling campaign (figure 4a), and the fitted occupancy by region and sampling campaign after the simple occupancy model testing the interaction between both factors (figure 4b).

Comment 33: Line 375 – other variables not only may, but definitely play a role in abundance variation.

Answer 33: Thank you for pointing this out, you are right, it has been modified.

Comment 34: Line 441 – “habitat variables were thoroughly examined” – I think there are many other habitat variables that may be considered - such as habitat fragmentation and connectivity or vegetation characteristics.

Answer 34: Thank you for pointing this out, you are right, it has been modified

Comment 35: Line 468 – “more detailed data on food availability, predator pressures” – actually this study does not include data on food availability or predator pressures at all.

Answer 35: Thank you for pointing this out, it has been modified 

Comment 36: Line 474 – “climatically marginal” – what does this mean?

Answer 36: It’s a mistake in the translation, it has been modified.

Supplementary Material

Comment 37: Table S1, S2 – Model instead of Modelo

Answer 37: Thank you for pointing this out, it has been modified.

Comment 38: Table S1-S4 – t instead of T

Answer 38: Thank you for pointing this out, it has been modified.

Comment 39: Table S3 - As there is no significant result here, I do not see the point in this table. One note in the caption of Fig. S2 would be enough.

Answer 39: Thank you for pointing this out, it has been modified, we erased the table and we mentioned it in Figure S2.

Comment 40: Table S5 – Aren’t these GLMMs? You should give more details in the caption on the response variable(s) and random factors, as well as the significance of R2m and R2c and of the values in the parantheses. „Model selection using the Dredge function first GLM model suggested” – this is not very clear. Please rephrase. What are the „*/////////” in the row for Log Likelihood? Akaike Inf, Crit, and Bayesian Inf, Crit, it’s . instead of ,  but best writein full.

Answer 40: thanks for pointing this out, this table has been moved to ‘Table 3’ as part of a suggestion from another reviewer. All corresponding explanations suggested in this comment have been added to table 3.

Comment 41: Table S6 – The title is very cryptic. What models? You should explain what the values in the parantheses mean and what Int stands for. Do not let the reader guess... It is Delta AIC not just Delta. The values for Weight are all 1 in this table. As these are the best competing models (or at least that is what I understand), this is not correct. Weights need to add up to 1.

Answer 41: thanks for pointing this out, this table corresponds now to Table S4 after the modifications. All corresponding explanations suggested in this comment have been added to table S4. And the issue with the weight has been fixed, it should be right now.

General answer: Thanks for the time and comments in order to improve our work, they have been of great help to enhance the quality of our article. 

Reviewer 3 Report

Comments and Suggestions for Authors

I am fine with the simple summary

Abstract I am fine with it – reads much smoother than the simple summary

The introduction is well written, describes the special local situation, the effect of global warming an change in land use, as well as the investigated species.

The research questions and hypothesis are well formulated

Fig. 1: Natura 2000. Is it necessary to indicate these areas: would be interesting to see the dimension of Pyrenees and Mediterranean on the map.

Study area and sampling design are well described as well as data analysis

Results

Line 269-276 Results are hard to follow. Maybe these results should be presented in a table

This is the same for change in land cover

The trapping results are very well presented in every aspect possible

I am fine with the description of the rest of the results

Discussion

The discussion is well written, has good rigor and covers all aspects quite well.

Conclusion I am fine with it

Author Response

Comments 5: Fig. 1: Natura 2000. Is it necessary to indicate these areas: would be interesting to see the dimension of Pyrenees and Mediterranean on the map.

Answer 5: The figure has been modified in such a way that the typo in Natura 2000 has been fixed, thank you for pointing this out. The areas ‘Mediterranean’ and ‘Pyrenees’ have been selected based on the study carried out by Sans Fuentes & Ventura in 2000, where they identified two main groups of bioregions according to Distribution Patterns of the Small Mammals (Insectivora and Rodentia).  

For the purposes of clarity and understanding of the map, we believe it is better not to add this differentiation, rather than the colourimetric identification of the sampled sites. Catalonia is divided into 13 physiographical regions (Figure 1, Sans-Fuentes & Ventura), which after performing a dendrogram of similarity between small mammal communities, two large groups were identified. Thus, we considered ‘Pyrenees’ to be the second branch in the first division, including the two regions Central and Eastern Pyrenees. And ‘Mediterranean’ sampling sites were associated to the group of the second division in the first branch, including Central segment of the southern pre-pyrenees, northern catalanidic territory, eastern auso-segarric territory, olositanic territory, western auso-segarric territory and eastern segment of the southern pre-pyrenees (Figure 2 Sans-Fuentes & Ventura). Clethrionomys glareolus belongs to the chorotype II after this classification, thus is not present in the first division of the first branch, matching with our findings.  

Comments 7: Line 269-276 Results are hard to follow. Maybe these results should be presented in a table

Comments 8: This is the same for change in land cover

Answer 7 and 8, thank you for identifying this issue, following your advice, we have tabulated the results for both climate and land use cover

Comments 1: I am fine with the simple summary

Comments 2: Abstract I am fine with it – reads much smoother than the simple summary

Comments 3: The introduction is well written, describes the special local situation, the effect of global warming an change in land use, as well as the investigated species.

Comments 4: The research questions and hypothesis are well formulated

Comments 6: Study area and sampling design are well described as well as data analysis

Comments 9: The trapping results are very well presented in every aspect possible

Comments 10: I am fine with the description of the rest of the results

Comments 11: The discussion is well written, has good rigor and covers all aspects quite well.

Comments 12: Conclusion I am fine with it

Regarding comments 1, 2, 3, 4, 6, 9, 10, 11, 12

Thank you for your appreciation of the work done in each section.

Round 2

Reviewer 1 Report

Comments and Suggestions for Authors

only few more comments, see attached

Author Response

Thank you for the time and comments in order to improve our work, we have addressed the minor comments, they have been of great help to improve the quality of our article. 

Reviewer 2 Report

Comments and Suggestions for Authors

Dear Authors, 

I have read the revised version of your manuscript and I consider my previous comments as satisfactorily addressed. I only have a few minor comments left, which are marked in the attached PDF.

Author Response

(The authors gave the same response as above.)
